# Evaluation of Diabetes Effects of Selenium Nanoparticles Synthesized from a Mixture of Luteolin and Diosmin on Streptozotocin-Induced Type 2 Diabetes in Mice

**DOI:** 10.3390/molecules27175642

**Published:** 2022-09-01

**Authors:** Rosa Martha Pérez Gutiérrez, Julio Téllez Gómez, Raúl Borja Urby, José G Contreras Soto, Héctor Romo Parra

**Affiliations:** 1Escuela Superior de Ingeniería Química e Industrias Extractivas, Instituto Politécnico Nacional (IPN), Unidad Profesional Adolfo López Mateos, Ciudad de México 07708, Mexico; 2Centro Anáhuac de Investigación en Psicología, Facultad de Psicología Universidad Anáhuac, Av. Universidad Anáhuac 46, Huixquilucan Edo., Ciudad de México 52786, Mexico; 3Lab. de Microscopía Electrónica de Transmisión, Centro de Nanociencias y Micro-Nanotecnologías (CNMN), Instituto Politécnico Nacional, Ciudad de México 07738, Mexico

**Keywords:** flavonoids, nanoparticles, selenium, diabetes, luteolin, diosmin

## Abstract

The absence of a treatment efficient in the control of type 2 diabetes mellitus requires more functional products to assist treatment. Luteolin (LU) and diosmin (DIO) have been known as bioactive molecules with potential for the treatment of diabetes. This work aimed to establish the role that a combination of LU and DIO in selenium nanoparticles (SeNPs) played in streptozotocin (STZ)- induced diabetes mice. Green synthesis of Se NPs was performed by mixing luteolin and diosmin with the solution of Na_2_SeO_3_ under continuous stirring conditions resulting in the flavonoids conjugated with SeNPs. The existence of flavonoids on the surface of SeNPs was confirmed by UV-Vis spectra, Fourier transform infrared spectroscopy (FTIR), transmission electron microscopy (TEM) images, and DLS graphs via Zetasizer. The average diameter of GA/LU/DIO-SeNPs was 47.84 nm with a PDI of −0.208, a zeta potential value of −17.6, a Se content of 21.5% with an encapsulation efficiency of flavonoids of 86.1%, and can be stabilized by gum Arabic for approximately 175 days without any aggregation and precipitation observed at this time. Furthermore, The C57BL/6 mice were treated with STZ induced-diabetes and were exposed to LU/DIO, SeNPs, and GA/LU/DIO-SeNPs for six weeks. The treatment by nanospheres (GA/LU/DIO-SeNPs) in the mice with diabetes for a period of 6 weeks restored their blood glucose, lipid profile, glycogen, glycosylated hemoglobin, and insulin levels. At the same time, there were significant changes in body weight, food intake, and water intake compared with the STZ- untreated induced diabetic mice. Moreover, the GA/LU/DIO-SeNPs showed good antioxidant activity examined by catalase (CAT), superoxide dismutase (SOD), glutathione peroxidase (GPx) in liver and kidney and can prevent the damage in the liver evaluated by aspartate aminotransferase (AST), alanine aminotransferase (ALT), and alkaline phosphatase (ALP) activities. The nanospheres exhibited a significant anti-diabetic activity with a synergistic effect between the selenium and flavonoids. This investigation provides novel SeNPs nanospheres prepared by a high-efficiency strategy for incorporating luteolin and diosmin to improve the efficiency in type 2 diabetes.

## 1. Introduction

Diabetes mellitus is a chronic metabolic disease that is principally characterized by inflammation, insulin resistance, hyperlipidemia, hyperglycemia, increased formation of reactive nitrogen (RNS) species, and reactive oxygen species (ROS). Chronic hyperglycemia reduces the antioxidant capacity of CAT, SOD, GPx, and erythrocyte glutathione (GSH), provoking insulin resistance through the inhibition of insulin signals [1]. In type 2 diabetes mellitus, the main cause of mortality and morbidity are premature macrovascular complications which, despite current treatments including healthy nutrition, daily physical activity, monitoring lipid profile and arterial pressure, and oral antihyperglycemic drugs (thiazolidinediones, sulphonylureas, biguanides, and 𝛼-glycosidase inhibitors) are related to adverse side effects or reduction in response after prolonged treatment [2]. These treatments are inadequate for most patients [3]; therefore, it is imperative to explore options in medicinal plants and nanomedicine to better manage the disease.

Numerous studies have shown that nanoparticles can adsorb, fix, and carriage other compounds due to their large surface area [4]. Currently, there is growing evidence of the effectiveness of the use of nanoparticles for type 2 diabetes mellitus (T2DM) management and prevention. Nanomedicine is an emerging form of therapy that focuses on alternative drug delivery and improvement of the treatment efficacy while reducing detrimental side effects to normal tissues [5].

Selenium is an essential element for humans and can ameliorate the activities of the selenoenzymes and antioxidant enzymes, including GPx [6], which is associated with the immune system, helping to enhance the immune response in both the innate and acquired immune systems [7]. Selenium demonstrated insulin-like activities when administered in rats stimulating glucose transport and insulin-sensitive cyclic adenosine monophosphate phosphodiesterase (cAMP-PDE) when incubated with rat adipocytes [8]. Nanoparticles of selenium have received great attention for their unique biological activities and low toxicity and can alleviate hyperlipidemia, hyperglycemia, and in STZ-induced diabetic rats, possibly by eliciting insulin-mimetic activity [9]. SeNPs establish an attractive platform to transport different drugs to the target of action [10] due to the good absorptive ability produced by the interaction between -C-N-, -COO, C=O, and -NH2, groups of proteins with the nanoparticles of Se [7].

Bioactive compounds isolated from herbal plants have been shown to have different pharmacological properties. Selected compounds from plants have been evaluated for their potential in the management of diabetes. Among these naturally occurring products, we find luteolin and diosmin. In this context, luteolin (LU; 3’,4’,5,7-tetrahydroxyflavone) is a bioflavonoid widely distributed in the plant kingdom and consists of a C6-C3-C6 structure that possesses a 2–3 carbon double bond and four hydroxyl groups at carbons 5, 7, 3′, and 4′ [11]. Luteolin, reduce neuroinflammation and ameliorate brain insulin resistance protecting against the development of Alzheimer’s disease [12]. Luteolin alleviates oxidative stress, neuroinflammation, and neuronal insulin resistance restoring blood adipocytokines levels in the mouse brain [13]. Another research demonstrated that LU significantly reduces insulin resistance, improves homeostatic model assessment for insulin resistance (HOMR-IR), glycosylated hemoglobin (HbA1c,) and reduces blood glucose levels [14]. LU treatment improved malondialdehyde (MDA) content, SOD activity, and the expression of the heme oxygenase-1 (HO-1) protein [15]. Diosmin, a bioflavonoid glycoside (diosmetin 7-rutinoside) found in citrus fruits, can stimulate the secretion of β-endorphin, associated with the improvement of hyperglycemia by regulating glucose homeostasis and increasing glucose uptake in soleus muscle and reducing hepatic gluconeogenesis [16]. Previous studies have reported a significant decrease in C-reactive protein (CRP), HbA1c, and levels of plasma lipids such as triglycerides (TG), low-density lipoprotein cholesterol (LDL), high-density lipoprotein cholesterol (HDL-CHO), very low-density lipoprotein cholesterol (VLDL-CHO), phospholipids, and free fatty acids [13]. Diosmin increased the activity of antioxidant enzymes, such as GPx, and SOD [17]. Diosmin possesses neuroprotective effects against long-term potentiation deficits and TBI-induced memory due to a decrease in the pro-inflammatory cytokine TNF-α level in the hippocampus [18].

The main objective of the present study is to design a new therapeutic strategy for type 2 diabetes where combined SeNPs with flavonoids (luteolin and diosmin) to investigate whether improve the therapeutic effect by a synergic effect on STZ-induced diabetes mellitus in C57BL/6 mice.

## 2. Results

### 2.1. Formation of SeNPs

The formation of SeNPs was observed by the solution color as well as the UV-vis spectra. Na_2_SeO_3_ reacted with ascorbic acid containing Arabic gum (GA), and the color of the solution was turned from colorless to orange-red, indicating the selenium nanoparticles formation (Figure 1A). The effect of Arabic gum (0.5–1%) as a stabilizer on SeNPs formation was investigated by monitoring the absorption spectra of the colloidal solutions. The nanoparticles showed a characteristic absorption peak at 270 nm (Figure 1B), associated with the surface plasmon resonance of SeNPs. The findings agree with another investigation on SeNPs synthesis [19]. The spectra of UV absorption indicated that flavonoids (Figure 1B) were successfully absorbed on the surface of SeNPs, as shown in Figure 1B, corresponding to the spectra of LU/DIO combination and GA/LU/DIO-SeNPs, respectively.

### 2.2. Fourier Transforms Infrared Spectroscopy (FTIR)

FTIR spectra were performed to characterize the interaction between Se nanoparticles and flavonoid molecules. Wavelength absorbance for GA/LU/DIO-SeNPs was observed between the range of 500–4000 cm^−1^. Figure 2A–C shows the spectra of SeNPs, LU/DIO mixture, and GA/LU/DIO-SeNPs, respectively. LU/DIO spectra indicated the presence of a band around 3300 cm^−1^ associated with the stretching vibration of the hydroxyl group (OH), mainly in phenolic groups [20]. While a peak near 1407 cm^–1^ and the bands around 798 cm^–1^ are due to aromatic groups (C-C stretch). Thus, the OH groups of GA-SeNPs have characteristic absorption peaks at 3414 cm^−1^, whereas in GA/LU/DIO-SeNPs, observed a band of OH groups around 3449 cm^−1^. The shift of the hydroxyl group peak indicates a strong bonding interaction between hydroxyl groups of GA, flavonoids, and surface atoms of SeNPs, leading to the stabilization of nanoparticles.

### 2.3. Characterization of GA/LU/DIO-SeNPs

Figure 3A shows a DLS analysis of the colloidal solution of GA/LU/DIO-SeNPs, indicating a size range of 20–118 nm with an average size of 47.84 nm. In the histogram, the frequency distribution displayed maximum synthesized particles were below 100 nm. The polydispersity index (PDI) observed was 0.208 indicating the low aggregation of particles. The DLS assay evaluated the hydrodynamic diameter in consequence, the size determined by DLS was observed to be greater than the size analyzed by TEM micrographs. The DLS analysis agreed with the TEM results. The zeta potential anticipates the stability of nanoparticles in a colloidal solution. The zeta potential value of −17.6 is shown in Figure 3B. The magnitude of the negative zeta potential value confirms lower aggregation and the higher stability of nanoparticles. Representative morphology of nanoparticles by transmission electron microscopy (TEM) images of GA/LU/DIO-SeNPs are shown in Figure 3C,D. It revealed that nanoparticles had a monodisperse, uniform, and spherical structure with a diameter of around 20–50 nm, which agreed with that of the GA/LU/DIO-SeNPs measured by DLS at 47.84 nm. TEM images support that the GA and SeNPs were packed closely to form spherical structures.

### 2.4. Analysis by Energy-Dispersive X-ray (EDX)

Furthermore, the surface elemental compositions of SeNPs and GA/LU/DIO-SeNPs were analyzed using dispersive X-ray energy (EDX (Figure 4A,B). Analysis of SeNPs mainly displayed the presence of three strong signals from the C atom (28.4%), Se atom signal (68.2%), and O atom signal (3.4%). It also showed two no distinct peaks of other elements. Findings indicated that elemental Se is present in SeNPs (Figure 4A). The surface elemental compositions of GA/LU/DIO-SeNPs also were evaluated using EDX assay (Figure 4B). EDX displayed three strong signals from the C atom (69.1%), together with an O atom signal (9.3%) Se atom signal (21.5%). EDX does not show other significant impurity peaks. Findings demonstrated that elemental Se is present in both nanoparticles SeNPs and GA/LU/DIO-SeNPs, while C and O elements may derive from the flavonoids. These results supported that GA/LU/DIO-SeNPs, establish a Se-flavonoid conjugation rate of 21.5%, demonstrating that the nanoparticles of selenium in the presence of flavonoids were successfully synthesized

### 2.5. Stability of GA/LU/DIO-SeNPs

As shown in Figure 4B, GA/LU/DIO-SeNPs remained stable over a storage period of 175 days without any aggregation and precipitation observed at this time.

### 2.6. In Vitro Release of Flavonoids

Luteolin and Diosmin combination release from nanoparticles was performed at pH of 5.5 and 7.4 at 37 °C, which was described with an initial fast release followed by a slower and gradual flavonoid release (Figure 4C).

### 2.7. Encapsulation Efficiency (EE)

EE indicates the amount of luteolin and diosmin encapsulated in the SeNPs. The EE of luteolin and diosmin mixture and Se were 86.1%.

### 2.8. Anti-Diabetic Effect of LU/DIO, SeNPs, and GA/LU/DIO-SeNPs on STZ-Induced Diabetes Mice

#### 2.8.1. Effect on Body Weight, Food Intake, and Water Intake in C57BL/6 Mice

To investigate the effect of LU/DIO, SeNPs, and GA/LU/DIO-SeNPs on the basic physiological indexes of mice, we analyzed body weight, food intake, and water intake in C57BL/6 mice during the 6-week study period. Results in Table 1 indicate that STZ caused a loss of body weight increase in average water and food consumption. The weight of the diabetic control group was decreased by 12%. After LU/DIO, SeNPs GA/LU/DIO-SeNPs, and Mtf treatment, the body weight gain was 17, 5.30, 32.6 and 12.4%, respectively, in diabetic mice, indicating that the weight of mice increased more slowly in the diabetic group. The average daily water intake (Table 2) and food intake (Table 3) in mice is significantly higher in DM mice. During the experimentation, it was observed that the water intake and food intake showed a trend of decrease to those in the DM mice and exhibited values near normalcy towards the normal control group. Results indicated that GA/LU/DIO-SeNPs showed better results than metformin on physiological indexes in diabetic mice.

#### 2.8.2. Oral Glucose Tolerance (OGTT) Test

The ability of the animals to tolerate glucose loading was examined using the oral glucose tolerance test, which gives information about insulin secretion following a meal observed in the development of T2DM and provides information on the effect of a drug in animals handling elevated blood glucose. Blood glucose levels show a significant change than glucose load, increasing rapidly in all diabetic groups within the first 30 min and remaining elevated for the next minutes (Table 4). Oral treatment of LU/DIO, SeNPs, GA/LU/DIO-SeNPs at 10 mg/kg in diabetic mice produced a significant (*p* < 0.05) decrease of 45.7, 28.8 and 54%, respectively, at 120 min after glucose administration. At the same time, glibenclamide (GBL; 5 mg/kg) showed a reduction in blood glucose of 35% at the same time. GA/LU/DIO-SeNPs exhibited an anti-diabetic effect at 120 min after the glucose load better than those observed in the GBL group (35%).

#### 2.8.3. Insulin Tolerance Test (ITT)

An insulin tolerance test was performed to determine insulin sensitivity in the presence of LU/DIO, SeNPs, and GA/LU/DIO-SeNPs. Results indicated that insulin resistance was severe in the diabetic group (Table 5). Administration of LU/DIO, SeNPs, GA/LU/DIO-SeNPs, and GBL decrease serum glucose levels by 58, 45, 63 and 54%, respectively. Findings are in accordance with our results on HOMA-IR index and insulin levels, indicating that GA/LU/DIO-SeNPs, for his response to insulin sensitivity and glucose levels, exerted an improvement against insulin resistance in C57BL/6 mice with a greater effect than those produced by LU/DIO, SeNPs, and GBL indicating that this effect can be for the synergic effect of selenium and flavonoids.

#### 2.8.4. Effect on Organ Weight

As shown in Table 6, no significant difference in the weight of the kidney was observed between the experimental groups. Table 6 displayed that the weight of the liver of the diabetic control group was elevated (37.5%) compared to those in the normal control group; this increase observed in liver weight could be due to increased cholesterol and lipid content. The liver weight of the group of diabetic mice treated with GA/LU/DIO-SeNPs decreased to 36.35%, returning to normal weight.

#### 2.8.5. Biochemical Parameters in C57BL/6 Mice

The results presented in Table 6 demonstrated that diabetic mice induced with STZ cause significant hypercholesteremia (T-CHO), and hypertriglyceridemia (TG). Low-density lipoprotein cholesterol (LDL-CHO), high-density lipoprotein cholesterol (HDL-CHO), and free fatty acid (NEFA). In addition, a significant increase in serum glucose, AST, ALT, ALP, HbA1c, and thiobarbituric reactive species (TBARS) was also observed, while HDL-c concentration, serum insulin level, CAT, SOD, and GPx exhibited a significant reduction when compared with the control group.

#### 2.8.6. Effect on the Fasting Blood Glucose and HbA1c

LU/DIO, SeNPs, and GA/LU/DIO-SeNPs at 10 mg/kg in 6 weeks of treatment in STZ diabetic mice exhibited significant anti-diabetic effect (Table 6), and significantly reduced altered blood glucose levels values in 40.53, 32.19 and 58.18%, respectively. The decrease in blood glucose levels in metformin-treated T2DM mice (52.7%) was comparable to that of the LU/DIO, and GA/LU/DIO-SeNPs. The Findings demonstrate that GA/LU/DIO-SeNPs showed maximum inhibition HbA1c of 52%, whereas LU/DIO, was 42.5%, SeNPs of 35.4%, and Mtf of 34% (Table 6).

#### 2.8.7. Hypolipidemic Effect in STZ-Induced Diabetic Mice

Concerning the lipid parameters, continuous administrations of LU/DIO, SeNPs, GA/LU/DIO-SeNPs for 6 weeks period, a significant reduction (Table 6) was observed in levels of T-CHO, TG, LDL-CHO, and NEFA in the serum. GA/LU/DIO-SeNPs turned out to be the most effective in alleviating these significant parameters in diabetic animals. Results demonstrated that LU/DIO, SeNPs, and GA/LU/DIO-SeNPs showed a decline in these parameters than that of the mice of the diabetic control group resulting enhance lipid metabolism disorder in C57BL/6 mice. On the other hand, a significant increase in HDL-CHO concentration of 47.5, 29.5 and 65%, respectively, was observed, with results comparable with metformin (46%).

#### 2.8.8. Improvement of Insulin Resistance and HOMA-IR

To evaluate insulin resistance, we determined OGTT, ITT, plasma insulin, and HOMA-IR. The serum insulin level was significantly decreased in the diabetic control group to those in the other experimental groups. The finding suggested that the function of insulin secretion of DM mice has been affected by continued hyperglycemia. Treatment of diabetic rats with LU/DIO, SeNPs, and GA/LU/DIO-SeNPs at 10 mg/kg for 6 weeks significantly increased the insulin level compared to the diabetic group (Table 6); of these treatments GA/LU/DIO-SeNPs mg/kg showed an increase in insulin secretion of 11.9% (38.8 mIU/L) which is nearer to the values exhibited by the normal control group (39.1 mIU/L). In addition, significantly (*p* < 0.05) reduced the levels of HOMA-I in T2DM mice by 30.76, 26.4 and 50.16%, respectively, compared with the DM group (Table 6).

#### 2.8.9. Nanoparticles Prevent Changes in Oxidative Stress Parameters in Diabetic Liver and Kidney

Oxidative stress of the liver and kidney were evaluated by measuring the activities of antioxidant enzymes and liver peroxidation. In the diabetic control group was observed a significant reduction level of the antioxidant enzymes and an increase in lipid peroxidation when compared to the normal control group.

In liver and kidney, treatment of LU/DIO, SeNPs, GA/LU/DIO-SeNPs, and Mtf for 6 weeks exhibited remarkable improvement in the SOD, GPx, and CAT. The SOD level significantly (*p* < 0.05) increased in the liver and kidney after treatment with GA/LU/DIO-SeNPs these results were found to be comparable to LU/DIO, SeNPs, and metformin-treated groups. While in T2DM mice, other oxidative stress parameters CAT and GPx levels with GA/LU/DIO-SeNPs displayed a significant (*p* < 0.05) increase in liver and kidney. Findings demonstrated that nanoparticles alleviated oxidative stress. In addition, malondialdehyde (MDA) levels in the liver and kidney (used as a marker for lipid peroxidation) are shown in Table 6. Results indicated that the STZ-induced diabetes mice showed a significant (*p* < 0.05) increase in MDA levels in the liver and kidney as compared to the normal group. Findings reveal that GA/LU/DIO-SeNPs showed a significant reduction in MDA levels in the liver (66.8%) and kidney (80.77%) were comparable to metformin (liver 65%; kidney 77.28%) demonstrated a decrease of lipid peroxidation in the diabetic mice. While GA/LU/DIO-SeNPs were found to be better in increasing the level of the antioxidant enzymes and in a decrease of lipid peroxidation among the treated groups.

#### 2.8.10. Effect on Serum Hepatic Functions Biomarkers

AST, ALT, and ALP which is considered an indication of the functional integrity loss and cellular leakage of the liver cell membrane [20]. Moreover, the changes in the liver functions in diabetic mice treated with LU/DIO, SeNPs, GA/LU/DIO-SeNPs, and Mtf for 6 weeks resulted in a considerable reduction in ALT, AST, and ALP activities in contrast to the diabetic groups, results are illustrated in Table 6. However, treatment with GA/LU/DIO-SeNPs caused a significant reduction in liver levels of ALT, AST, and ALP by 45.4, 45 and 67.5%, respectively, as compared to the other groups. This treatment showed a significant reduction compared to metformin. Findings support the synergistic effect of flavonoids with selenium.

#### 2.8.11. Effect on Liver Glycogen Levels

As demonstrated in Table 4, glycogen levels in the liver were significantly depleted in diabetic mice when compared with the control. The depleted liver glycogen content was, however, reversed in LU/DIO, SeNPs, GA/LU/DIO-SeNPs, and Mtf groups with 50, 17, 68 and 47%, respectively, compared with the diabetic group.

## 3. Discussion

Selenium nanoparticles (SeNPs) were prepared by using ascorbic acid as reducer and gum Arabic (GA) as the stabilizer in a facile synthetic approach where selenium oxyanion (SeO_3_^2^) in an aqueous medium which when reacting with the ascorbic acid, is reduced to elemental selenium (Se^0^). The zeta potential is an electrokinetic potential associated with the stability of nanosystems used to determine the stability of nanoparticles [21]. The zeta potential of the synthesized SeNPs and GA/LU/DIO-SeNPs indicated its stability in colloidal solution, suggesting that the repulsion force of bare SeNPs was high and thus much more difficult to aggregate in an aqueous solution. The results demonstrated that nanoparticles did not precipitate over a storage period of 175 days, confirming their stability. The SeNPs without GA precipitates, lost transparency after seven days, supporting that the addition of Arabic gum enhances the stability of the GA/LU/DIO-SeNPs, causing the nanoparticles to remain dispersed during the storage period. The negative charge on the surface of the NPs indicates major stability of gum-stabilized nanoparticles due to the intrinsic capping nanoparticles by proteins and polysaccharides of the gum [22]. This was verified by utilizing TEM and DLS analysis. The morphology of SeNPs and GA/LU/DIO-SeNPs in TEM analysis showed spherical particles, monodisperse and homogeneous with a relatively smaller diameter. This phenomenon can be due to GA changing the interface of NPs controlling their growth and stabilizing the nanoparticles solution due to their active hydroxyl groups and complex branch structures, indicating that GA, which was adsorbed on the surface of SeNPs, prevents their aggregation.

The nanoparticle solutions showed characteristic absorption peaks in UV/vis spectrum at 270 nm, associated with surface plasmon resonance of SeNPs. These observations agree with previous investigations on spherical SeNPs synthesis [23]. FTIR analysis was performed to evaluate the interaction between SeNPs and flavonoid molecules. The spectrum of GA/LU/DIO-SeNPs provides evidence that GA and LU/DIO form part of the nanoparticle.

The highest entrapment of 86% suggested a relatively large affinity between luteolin and diosmin and the nanoparticle matrix. Encapsulation efficiency could be ascribed to the accessibility and amount of potential binding sites among the flavonoids and Se. The synthesis process, particle size, and encapsulation reaction are some factors that affect the EE, as well as luteolin and diosmin as core materials also contribute to the EE loaded in SeNPs. In addition, the smaller particle size of GA/LU/DIO-SeNPs with a greater specific surface area and stronger adhesion capacity participated in the greater encapsulation efficiency of 86.1%.

The in vitro release of the drug is considered an important factor in the prediction of drug bioavailability from different formulations. Within the first 60 min, the burst release phase was predominantly caused by adsorbed, or desorption of flavonoids surface bound. In the second phase, the release was slower, associated with nanoparticle matrix degradation or erosion, the channels of the nanoparticles, and diffusion of the flavonoid through the pores [24].

Diabetes is characterized by the loss or degradation of structural proteins exhibiting a severe loss in body weight caused by proteolysis and an insulin deficiency triggering a reduction in the protein content in the muscular tissue [25]. GA/LU/DIO-SeNPs prevent loss of weight in diabetic mice group suggesting that they may reduce glycogenolysis, gluconeogenesis, and proteolysis to ameliorate the level of insulin [25]. In contrast, food intake reduction is associated with neuronal pathways and complex hormonal involving satiety modulations and appetite [25]. Results indicated that food intake reduction simply reduces energy intake which eventually lowers fat mass and blood glucose [26]. GA/LU/DIO-SeNPs can regulate water consumption in T2DM mice and alleviates hyperglycemia symptoms.

In the OGTT, was observed an improvement in blood glucose levels in mice treated with nanoparticles as compared to the control group. Treatment with GA/LU/DIO-SeNPs noticeably ameliorated impaired glucose tolerance by enhancing insulin resistance as well as the increase in peripheral utilization of glucose [27]. Results suggest that GA/LU/DIO-SeNPs can exert a hypoglycemic effect through the insulin-like effect of improving glycogenesis and enhancing cellular uptake of glucose compared to the GBL (sulfonylurea), which activates the release of insulin from the beta cells of the pancreas [28]. Results agree with previous investigations indicating that many flavonoids produced stimulation of insulin release and, in consequence, exert a hypoglycemic effect [29]. Results indicated that nanospheres treatment obviously alleviated insulin sensitivity observed in insulin tolerance tests, subsequently promoting glucose metabolism and declining fasting blood glucose levels.

The glucose peak in nondiabetic groups in the experiment maintained normal blood glucose levels, while diabetic animals showed a severe hyperglycemic effect throughout the experimental period. GA/LU/DIO-SeNPs treatment showed significantly fasting blood glucose levels compared with diabetic mice, and blood glucose returned to baseline levels. Findings suggested a significant recovery in the plasma insulin level.

Hemoglobin is the protein that carries oxygen to different cells of the organism. An increased level of glucose in the blood leads to its binding to hemoglobin generating glycated hemoglobin or causing hemolysis of red blood cells by increased lipid peroxide [30] associated with the production of ROS related to hyperglycemia causes [31]. Administration of LU/DIO, SeNPs, and GA/LU/DIO-SeNPs reduced HbA1c concentration compared to the untreated diabetic mice. These activities may be associated with the antioxidant effects and their capacity to reduce lipid peroxidation of these treatments, as was demonstrated from this investigation results.

Generally, diabetes is associated with dyslipidemia due to the insufficient or absence of insulin which participates in the regulation of lipid metabolism. Previous investigations have demonstrated that hypertriglyceridemia in diabetes stimulates a change in the oxidant-antioxidant balance and, consequently, increases lipid peroxidation [32].

In streptozotocin-induced diabetic mice, hypercholesterolemia was observed with an increased level of triglycerides, total cholesterol, and LDL-cholesterol due to an increase in cholesterol biosynthesis and intestinal absorption [33]. Treatment with nanoparticles decreases triglycerides, total cholesterol, and LDL-cholesterol to a significant level (*p* < 0.05); in addition, it produced a significant increase in HDL levels compared with nondiabetic mice. Results demonstrated that NPs have hypocholesterolemic properties due to improving lipid profiles, also causing a decrease in antioxidant enzymes, and protecting the damage to the liver and kidney. An imbalance in the body’s redox homeostasis produces oxidative stress causing damage to cell structures, including nucleic acids, proteins, carbohydrates, and lipids which is an improvement by nanospheres reestablishing the level of lipid profiles towards the normal group.

The administration of STZ to mice causes deterioration in the activities of antioxidant-related enzymes such as CAT, SOD, and GPx. When the activities of these antioxidant enzymes are reduced by the excess of hydrogen peroxide radical and superoxide anion, these stimulate ROS production, and lipid peroxidation. Thus, ROS cause the partial destruction of β-cells playing an important role in the pathology of diabetes mellitus. Supplementation of GA/LU/DIO-SeNPs in STZ-induced diabetic mice to increase the activities of CAT, GPx, and SOD in the kidney and liver. It also showed a significant reduction in kidney and liver MDA levels compared to the diabetic control. The improvement of oxidative stress in diabetic mice may be due to the synergistic ability of antioxidant and anti-diabetic of luteolin, diosmin, and SeNPs together in GA/LU/DIO-SeNPs.

The current research demonstrated liver damage by evaluating AST, ALT, and ALP enzymes. A raised level of these markers in diabetic mice indicated the injury of hepatocytes [34]. Thus ALT, and AST, liver enzymes, are found in the blood after hepatic cell injury. ALT is more specific to liver damage, while AST and ALP generally are associated with liver injury in cases of muscle damage and cardiac infarction; both markers are linked to hepatic cell function [35]. Elevation of oxidative stress in the liver is the cause of the increased release of liver markers, AST, ALT, and ALP due to injured hepatocytes. Six weeks of GA/LU/DIO-SeNPs treatment provokes a significant reduction in liver enzyme levels as compared to the diabetic control, suggesting a hepatoprotective effect, repairing liver cells, decreasing liver enzyme leakage into the blood, and reestablishing the level of liver markers towards the normal group.

It has been reported that insulin resistance decreases the capacity of hepatic tissue to synthesize glycogen and utilizes and decomposes glucose; consequently, more glucose is secreted into the blood. Our results demonstrated that GA/LU/DIO-SeNPs could increase hepatic glycogen content in mice, which implied that nanospheres could regulate hepatic glucose. In diabetic conditions, the synthesis of hepatic glycogen content is decreased due to the low level of insulin, which inactivates the glycogen synthase pathway [36]. The restoration of hepatic glycogen content by the GA/LU/DIO-SeNPs group demonstrated that the combination of flavonoids and Se stimulates the glycogen synthase enzyme by increasing insulin generation from pancreatic cells.

## 4. Materials and Methods

### 4.1. Materials

All the chemicals were purchased from Sigma Co.Ltd. (St. Louis, MO, USA). All the solvents in this study were of analytical grade, and aqueous solutions were prepared using Milli-Q water.

### 4.2. Synthesis of GA/LU/DIO-SeNPs

Briefly, 0.5% Arabic gum (GA) solution was added dropwise into 10 mM Na_2_SeO_3_ solution, and the mixture was diluted to 100 mL with Milli-Q water. The solution was mixed with 2 mg/mL of a combination of luteolin (LU) and diosmin (DIO; 1:1) under magnetic stirring. After, 40 mM ascorbic acid as the reducing agent [37] was mixed and kept stirring for 24 h in the dark at 37 °C. The obtained mixture of GA/LU/DIO-SeNPs was dialyzed with a dialysis bag (Mw cut-off: 3 kDa) in Milli-Q water to remove the excess sodium selenite. The GA/LU/DIO-SeNPs were collected by centrifugation at 15,000 rpm at 15 min and were washed twice with H_2_O, Milli-Q.

### 4.3. Characterization of GA/LU/DIO-SeNPs

#### 4.3.1. Zeta Potential and Particle Size Analysis

The zeta potential of nanoparticles and particle size were measured using laser Doppler velocimetry (LDV, Zetasizer Nano ZS, Malvern, UK). Between 1.0 and 1.5 mL of each sample was measured in a polystyrene cuvette at a fixed angle of 173° at 25 °C to obtain zeta potential values to check the bonding mechanism. The non-invasive backscattering (NIBS) assay was used to determine the particle size distribution. The results are reported as the average values of triplicate determinations. Dynamic light scattering (DLS) was used to monitor particle size changes during storage in a Zetasizer Nano ZSP, Malvern Instruments Ltd., Malvern, UK.

#### 4.3.2. Transmission Electron Microscopy (TEM)

The morphology of GA/LU/DIO-SeNPs was observed by TEM (Model HT7700, Hitachi High-Tech, Tokyo, Japan). NPs solution was prepared by sonic oscillation and added directly to carbon-coated copper grids to evaporate the solvent. The micrographs of NPs were then analyzed at an accelerating voltage of 80 kV.

#### 4.3.3. FT-IR Analysis

The FT-IR spectra of LU/DIO, SNPs, and GA/LU/DIO-SeNPs were characterized using a spectrophotometer FTIR-TENSOR27, Bruker, Ettinger, Germany, in the range of 400–4000 cm^−1^.

#### 4.3.4. UV-Visible Analysis

The UV-visible spectra of the LU/DIO, SNPs, and GA/LU/DIO-SeNPs solutions were obtained using a UV-250 Shimadzu spectrophotometer, Tokyo, Japan. The absorbance was determined in the range of 250–700 nm.

#### 4.3.5. EDX Analysis

The elemental compositions of the GA/LU/DIO-SeNPs were measured using energy-dispersive X-ray spectroscopy (EDS) by SEM-EDX spectrometer (Oxford, UK).

#### 4.3.6. Encapsulation Efficiency

Encapsulation efficiency was measured by quantifying the difference between the total amount of flavonoids used for the preparation of NPs and the amount of free flavonoids in the nanosuspension. Briefly, 2 mL of GA/LU/DIO-SeNPs preparation was centrifuged at 15,000 rpm for 45 min. The flavonoid content in the supernatant was measured by using UV spectrophotometer at 280 nm and determined from the mixture of luteolin and diosmin (LUDIO) standard curve. The encapsulation efficiency of flavonoids was calculated using Equation (1):(1)$$\mathrm{EE}\left(\%\right)=\frac{\mathrm{TOTAL}\;\mathrm{LUDIO}\;-\;\mathrm{LUDIO}\;\mathrm{supernatant}\;}{\mathrm{Total}\;\mathrm{Luido}}$$

#### 4.3.7. In Vitro Release of LU/DIO

Flavonoids-loaded NPs equivalent to 5 mg (LU/DIO) were added with 5 mL of phosphate-buffered saline pH 7.4 (donor medium), then it was explored in a dialysis bag (3500Da, Thermo Scientific, Rockford, IL, USA), immersed in 50 mL of pH 7.4 phosphate buffer which simulated intestinal fluid (SIF) and simulated blood fluid (SBF) and incubated at 37 °C for 30 h. After, at time intervals of 10, 20, 30 and 40 h, aliquots of 1 mL of the sample were used to measure LU/DIO release. The same analysis was performed at pH 5.5 phosphate buffer simulated gastric fluid (SGF; receptor medium). The flavonoid concentration of the aliquots was measured at 280 nm.

The cumulative releasing of flavonoids was calculated using Equation (2):(2)$$\mathrm{Cumulative}\;\mathrm{releasing}\;\left(\%\right)=\frac{E}{E_{0}}\times 100$$

E: the amount of LU/DIO measured in the release medium

E_0_: the initial amount of LU/DIO in the NPs

### 4.4. Stability of the Nanoparticles

The stability of nanoparticles was evaluated as follows: 500 mg of GA/LU/DIO-SeNPs was suspended in 5 mL of phosphate-buffered saline (PBS) and maintained at 37 °C. The shelf life of the nanoparticles was recorded in pre-determined time intervals (every 8 days) using absorbance intensities of the solutions over 5.5 months.

### 4.5. Experimental Animals

One hundred and forty-four adult male C57BL/6 mice (10 weeks of age) used for this study were obtained from the Central Animal House of the Universidad Anahuac, weighing between 25 to 30 g. The mice were housed in groups of five per cage under a normal 12 h light/dark cycle, controlled humidity (50 ± 5%) and temperature (22 ± 2 °C), and acclimatized to laboratory conditions for one week before starting the experiments, mice had free access to food and water ad libitum throughout the experimental period. The experiments reported in this study followed the guidelines stated in “Principles of Laboratory Animal Care” (NIH publications 85-23, revised 1985) and Mexican Official Normativity (NOM-062-Z00-1999). In addition, all animal experiments were approved by the Animal Experiment Committee of Escuela Nacional de Ciencias Biológicas, IPN (Code: Folio ENCB/CEI/046/2021).

### 4.6. Induction of Type 2 Diabetic Mouse

Diabetes was induced at 10 weeks of age in males by intraperitoneal injection (i.p.) of a freshly prepared solution of streptozotocin in citrate buffer (40 mg/kg body weight [BW] for 5 consecutive days). Seven days after i.p. injection, blood samples were collected from the tail vein, and blood glucose levels were measured using a glucometer (One Touch Glucometer, Roche, Basel, Switzerland). The mice with blood glucose levels higher than 13.9 mmol/l were considered diabetic. To prevent death from hypoglycemic shock induced by streptozotocin, a 5% glucose solution was administered for 4 days. Supplementation of NPs was carried out for additional 6 weeks, and Metformin (Mtf) was used as a positive control.

### 4.7. Animal Grouping and Treatment Schedule

Forty-eight C57BL/6 mice were divided into the following 6 groups (*n* = 8): Group 1: normal control (received distilled); Group 2: STZ diabetic control (received distilled); Group 3: STZ + LU/DIO (10 mg/kg/day); Group 4: STZ + SeNPs (10 mg/kg/day); Group 5: STZ + GA/LU/DIO-SeNPs (10 mg/kg); Group 6: STZ + Metformin (Mtf; 150 mg/kg/day served as positive control). The different concentrations of the samples and Mtf were given daily orally through a cannula.

### 4.8. Monitoring of Body Weight, Food Intake, and Water Intake

The body weights of mice in different groups were measured and recorded every week. Similarly, changes in the amount of water and food consumed were monitored.

### 4.9. Biochemical Analyses of Serum of Diabetic Mouse

#### 4.9.1. Assessment of Insulin Resistance, Insulin, Glycated Hemoglobin, and Glucose in Serum

Blood samples were taken from the inferior vena cava and centrifuged at 4000 r/min for 10 min. Blood glucose was determined by the Accu Chek glucometer method (Roche, Basel, Schweizer). The serum insulin levels were detected using a mouse ELISA kit (Thermo-Fisher Scientific, Waltham, MA, USA). The insulin resistance (HOMA-IR) assessment was calculated with reference to existing literature formulas [38]. HbA1c was determined by ELISA kit assay.

#### 4.9.2. Determination of Lipid Profile

After completion of the experiment, mouse blood was collected, and plasma was separated by a centrifuge at a speed of 13,000 rpm for 10 min (4 °C). For lipid profile analysis of mice in all groups, T-CHO, TG, HDL-CHO, LDL-CHO, and free fatty acid (FFA) were measured using commercial assay kits according to the manufacturer’s indications (Sigma-Aldrich, St Louis, MI, USA).

#### 4.9.3. Liver and Kidney Weight Determination

At the end of the treatment period (6 weeks), the mice were fasted overnight, after which they were sacrificed by cervical dislocation and necropsied. Then liver, and kidney, were removed surgically.

#### 4.9.4. Determination of ALT, AST, ALP, MDA, Antioxidant Enzymes, and Glycogen in the Liver of Mice

Liver tissue was homogenized in PBS to prepare 10% liver homogenate, then centrifugated (4 °C, 3000 rpm, 10 min), and the supernatant was collected. The content of MDA, SOD, GPx, CAT, AST, ALT, and ALP in the supernatant of liver homogenate was determined according to the requirements of the instructions provided in ELISA kit methods (Ela science, Boston, MA, USA).

Liver glycogen content was measured using sodium acetate buffer (100 μL; pH 6), 5 μL of amyloglucosidase, and 20 μL of the homogenate to a final volume of 0.5 mL with double distilled water and incubated at 50 °C for 30 min. A control with everything except amyloglucosidase was also analyzed. A 300 μL aliquot of each sample was then added to 1 mL of glucose oxidase/peroxidase reagent (GOPOD, Megazyme) and incubated at 50 °C for 30 min. The glycogen content was calculated at 510 nm based on a calibration curve constructed by reacting _D_-glucose of various concentrations with the same GOPOD reagent [38].

MDA level was expressed as µmol/mg of protein, GSH-px level was expressed as U mol/g of protein, and SOD activity was expressed as U/mg of protein. CAT activity was expressed as U/mg of protein, and glycogen was expressed as mg/g of tissue.

#### 4.9.5. Determination of Production, MDA, and Antioxidant Enzymes in the Kidney of Mice

Kidney tissue was homogenized in PBS to prepare 10% kidney homogenate, after centrifugated (4 °C, 3000 rpm, 10 min), and the supernatant was collected. The production of MDA, the levels of CAT, SOD, and GPx, in the supernatant of kidney homogenate were measured according to the requirements of the instructions provided in reagent kits.

#### 4.9.6. Oral Glucose Tolerance Test (OGTT) and Insulin Tolerance Test (ITT) on Diabetic Mice

Oral glucose tolerance tests were performed by the following procedure: A total of 48 male 48 C57BL/6 mice were randomly divided into 6 groups of 8 mice in each group:

Group 1 Normal mice (1% Tween 80 in water, 10 mL/kg body weight)

Group 2 Diabetic control (treated with Streptozotocin (STZ)

Group 3 Diabetic + LU/DIO (10 mg/kg body weight)

Group 4 Diabetic + SeNPs (10 mg/kg body weight)

Group 5 Diabetic + GA/LU/DIO-SeNPs (10 mg/kg body weight)

Group 6 Diabetic control + Glibenclamide (5 mg/kg body weight)

LU/DIO, SeNPs, and GA/LU/DIO-SeNPs were orally administered to fasted mice following a period of one hour, all mice were orally administered 2 g glucose/kg of body weight and the blood was sampled at different time-points 30, 60, 90 and 120 min after the glucose administration [39]. Blood glucose levels were measured by the glucose oxidase method. The same fasting protocol used for OGTT was adopted for the determined insulin tolerance test (ITT). The samples were orally administered to fasted mice for a period of one hour; then, diabetic mice were injected intraperitoneally with insulin (2 U/kg). The glucose level was determined according to the technique previously described at 30, 60, 90 and 120 min after the insulin injection. Blood glucose levels were measured by the glucose oxidase method.

### 4.10. Statistical Analysis

The results are expressed as the mean ± S.D. (*n* = 8). One-way analysis of variance (ANOVA) was continued by honest significant difference (Tukey’s HSD) post hoc test for multiple comparisons; data were analyzed using GraphPad Prism 5 software (Version 5.0, San Diego, CA, USA). Post hoc assaying was carried out for intergroup comparisons using the least significant difference test. Statistical significance is displayed as follows: ^a^
*p* < 0.001, ^b^
*p* < 0.01, ^c^
*p* < 0.05 and * *p* < 0.001, ** *p* < 0.01.

## 5. Conclusions

The results of the present investigation demonstrated that Se-NPs synthesized with Luteolin and Diosmin (GA/LU/DIO-SeNPs) administration to STZ-induced diabetic mice provided an effective treatment against hyperlipidemia, hyperglycemia, and hepato-renal dysfunction; thus, these nanoparticles were able to ameliorate the related serum biochemical parameters improving glycemic control, decrease lipotoxicity preserve β-cell function, and in consequence insulin resistance. Since most flavonoids possess poor oral delivery to the target sites, these nanoparticles would increase drug retention time, increase drug uptake, and facilitate drug targeting offering protective functions through the modulation of different pathways. Our findings demonstrated that the synthesized nanoparticles have good potential to reduce the disorders of diabetes mellitus and can be effectively used to treat diabetic conditions.

## Figures and Tables

**Figure 1 molecules-27-05642-f001:**
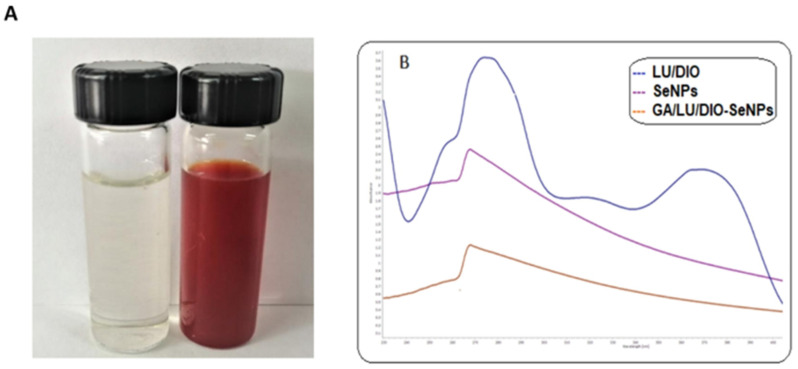
(**A**) formation of SeNPs synthesized with ascorbic acid and stabilizer with Arabic gum; (**B**) UV-visible spectrum for selenium nanoparticles (SeNPs), LU/DIO combination, and GA/LU/DIO-Se from 200 to 800 nm.

**Figure 2 molecules-27-05642-f002:**
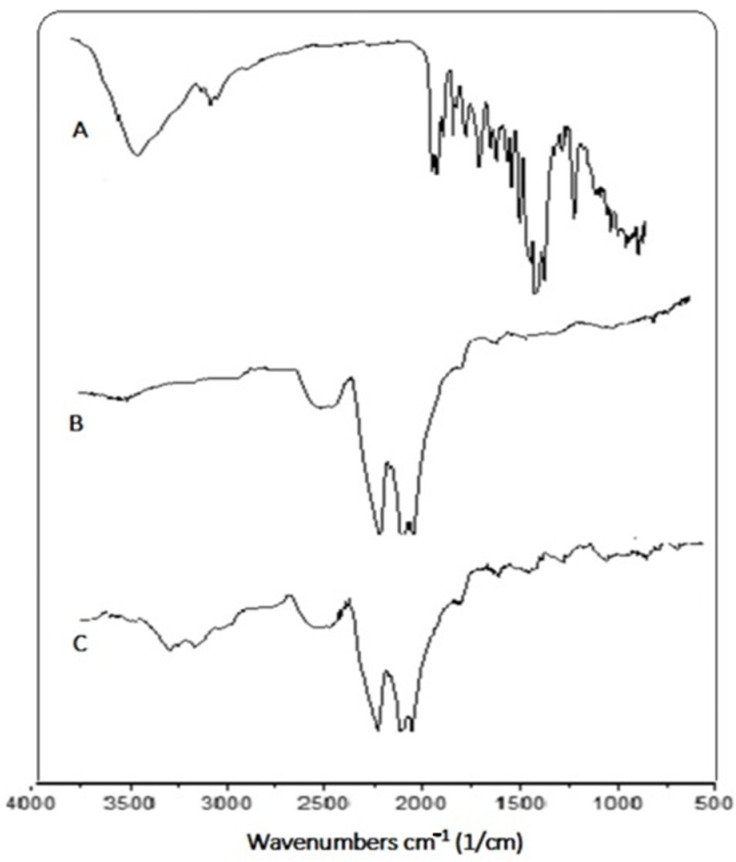
FTIR spectra: (**A**) LU/DIO mixture; (**B**) SeNPs; (**C**) GA/LU/DIO-SeNPs.

**Figure 3 molecules-27-05642-f003:**
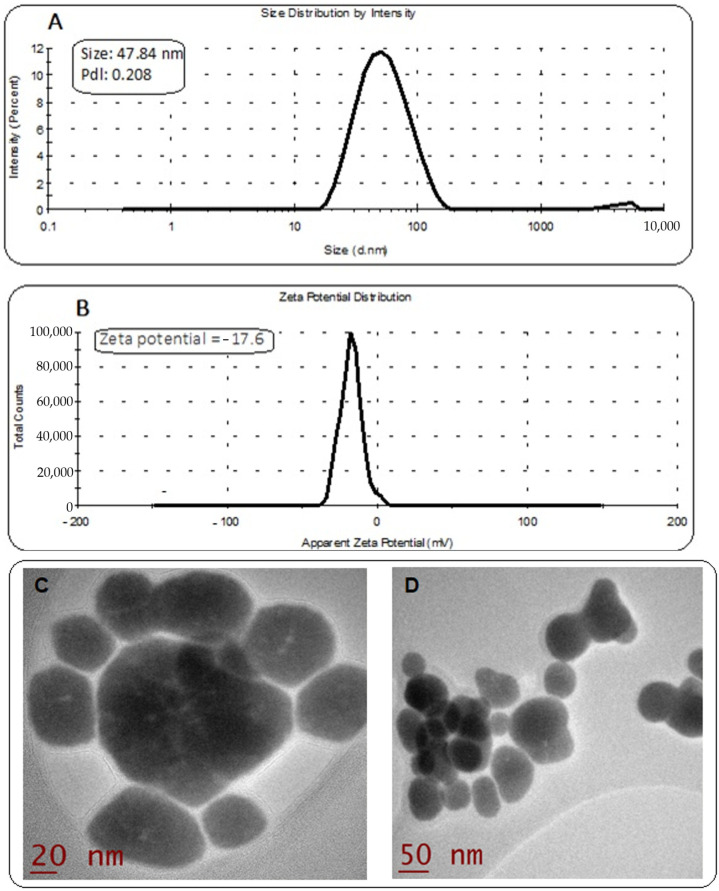
DLS analysis of synthesized GA/LU/DIO-SeNPs; size: 43–121 nm; average size: 70.9 nm (**A**) and zeta potential measurement (**B**); transmission electron microscope (TEM) images of LU/DIO-SeNPs in the presence of GA aqueous solutions (**C**,**D**).

**Figure 4 molecules-27-05642-f004:**
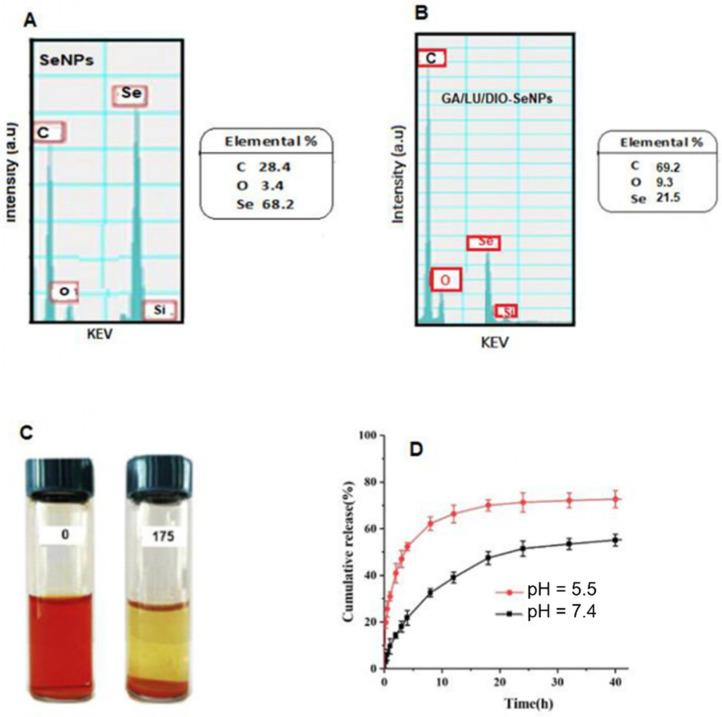
The surface elemental compositions by energy-dispersive X-ray (EDX) of SeNPs (**A**) and GA/LU/DIO-SeNPs (**B**); photographs of GA/LU/DIO-SeNPs during a 175-day period (**C**); cumulative percentage (%) release profile of flavonoids in GA/LU/DIO-SeNPs at pH = 5.5 and pH = 7.4 until to 40 h at 37 °C (**D**).

**Table 1 molecules-27-05642-t001:** Effect of flavonoid formulation (LU/DIO), SeNPs, and GA/LU/DIO-SeNPs on weight, in diabetic mice.

Parameters	Weight (g)						
	0 Days	1 Weeks	2 Weeks	3 Weeks	4 Weeks	5 Weeks	6 Weeks
Normal control	25.4 ± 2.3	26.0 ± 3.5	27.0 ± 2.25	27.6 ± 4.52	28.5 ± 5.20	29.2 ± 3.19	30.6 ± 4.50
Diabetic control	30.1 ± 4.02	29.2 ± 3.12 ^a^	28.6 ± 4.32 ^b^	28.0 ± 2.43 ^a^	27.0 ± 4.36 ^c^	26.8 ± 4.65 ^a^	26.5 ± 1.32 ^c^
DM + LU/DIO	24.7 ± 3.10	25.1 ± 5.10 *	25.8 ± 4.32 **	26.2 ± 3.19 *	27.4 ± 2.97 **	28.0 ± 5.53 *	28.9 ± 1.89 *
DM + SeNPs	26.4 ± 1.56	26.5 ± 2.65 **	26.6 ± 1.87 *	26.9 ± 3.19 **	27.4 ± 4.70	27.8 ± 2.67	27.8 ± 2.81
DM + GA/LU/DIO-SeNPs	24.2 ± 3.45	26.2 ± 3.4 * ^#^	29.3 ± 5.02 ** ^#^	30.4 ± 1.86 * ^#^	31.2 ± 3.93 * ^#^	31.7 ± 3.80 ** ^#^	32.1 ± 1.75 * ^#^
DM + Mtf	28.2 ± 2.29	28.7 ± 1.48 ** ^#^	29.3 ± 2.76 * ^#^	29.6 ± 5.19 ** ^#^	30.3 ± 0.49 ** ^#^	30.4 ± 1.85 * ^#^	31.7 ± 4.02 ** ^#^

Data are expressed as mean ± SD. * *p* < 0.05 and, ** *p* < 0.01 vs. diabetes group; ^a^
*p* < 0.05, ^b^
*p* < 0.01 and, **^c^**
*p* < 0.001 vs. control group (*n* = 8 for each group). ^#^
*p* < 0.05 vs. SeNPs group.

**Table 2 molecules-27-05642-t002:** Effect of flavonoid formulation (LU/DIO), SeNPs, and GA/LU/DIO-SeNPs on water intake in diabetic mice.

Parameters	Water Intake (mL/day)						
	0 Days	1 Weeks	2 Weeks	3 Weeks	4 Weeks	5 Weeks	6 Weeks
Normal control	8.3 ± 1.25	8.4 ± 0.94	8.6 ± 1.35	9.0 ± 2.01	9.1 ± 0.68	9.2 ± 1.16	9.3 ± 2.11
Diabetic control	8.6 ± 1.87	9.2 ± 1.24 ^a^	10.5 ± 2.17 ^b^	11.0 ± 1.91 ^a^	11.5 ± 3.12 ^c^	12.0 ± 2.54 ^a^	12.6 ± 2.34 ^b^
DM + LU/DIO	8.4 ± 2.16	8.6 ± 0.94 *	9.0 ± 1.42 **	9.2 ± 1.19 **	9.8 ± 2.97 **	10.0 ± 1.36 *	10.2 ± 1.42 **
DM + SeNPs	8.0 ± 0.98	9.0 ± 1.24	9.6 ± 1.87 *	10.0 ± 2.34 *	10.2 ± 0.78 **	10.8 ± 2.18 *	11.0 ± 3.02 *
DM + GA/LU/DIO-SeNPs	8.2 ± 1.56	8.6 ± 0.78 *	9.3 ± 0.85 *	9.4 ± 1.46 **	9.2 ± 1.20 ** ^#^	9.2 ± 0.87 ** ^#^	9.4 ± 2.04 * ^#^
DM + Mtf	8.1 ± 1.78	9.4 ± 1.21 **	9.6 ± 0.76 *	10.0 ± 2.12 *	10.3 ± 0.69 *	10.4 ± 1.32 *	10.6 ± 1.39 **

Data are expressed as mean ± SD. * *p* < 0.05 and, ** *p* < 0.01 vs. diabetes group; ^a^
*p* < 0.05, ^b^
*p* < 0.01 and, ^c^
*p* < 0.001 vs. control group (*n* = 8 for each group). # *p* < 0.05 vs. SeNPs group.

**Table 3 molecules-27-05642-t003:** Effect of flavonoid formulation (LU/DIO), SeNPs, and GA/LU/DIO-SeNPs on food intake, in diabetic mice.

Parameters	Food Intake (g/day)						
	0 Days	1 Weeks	2 Weeks	3 Weeks	4 Weeks	5 Weeks	6 Weeks
Normal control	5.8 ± 0.91	5.7 ± 0.67	5.9 ± 0.76	6.0 ± 0.53	5.9 ± 0.12	6.1 ± 0.43	6.1 ± 0.82
Diabetic control	6.1 ± 0.76	7.0 ± 0.84 ^a^	7.52 ± 1.62 ^c^	8.4 ± 1.12 ^a^	8.5 ± 1.04 ^c^	8.8 ± 0.75 ^b^	9.1 ± 0.65 ^a^
DM + LU/DIO	5.6 ± 0.56	5.7 ± 1.13 **	5.8 ± 4.32 *	6.2 ± 1.19 *	6.6 ± 0.95 *	6.8 ± 0.43 **	7.0 ± 0.53 *
DM + SeNPs	5.4 ± 0.71	6.2 ± 0.85 *	6.6 ± 1.87 **	6.9 ± 3.19 *	7.4 ± 0.77 *	7.8 ± 1.61 *	7.9 ± 0.42 **
DM + GA/LU/DIO-SeNPs	5.9 ± 0.89	5.6 ± 1.06 **	5.8 ± 1.03 **	6.1 ± 1.26 **	6.0 ± 0.95 ** ^#^	6.1 ± 1.20 ** ^#^	6.3 ± 0.80 ** ^#^
DM + Mtf	5.7 ± 0.49	5.8 ± 1.42 *	5.9 ± 1.46 *	6.2 ± 0.79 *	6.3 ± 0.58 ** ^#^	6.4 ± 1.85 * ^#^	6.5 ± 0.57 * ^#^

Data are expressed as mean ± SD. * *p* < 0.05 and, ** *p* < 0.01 vs. diabetes group; ^a^
*p* < 0.05, ^b^
*p* < 0.01 and ^c^
*p* < 0.001 vs. control group (*n* = 8 for each group). ^#^
*p* < 0.05 vs. SeNPs group.

**Table 4 molecules-27-05642-t004:** Effect of LU/DIO, SeNPs GA/LU/DIO-SeNPs and on oral glucose tolerance assay (OGTTin diabetic mice.

Groups	Plasma Glucose (mg/dl)				
	0 Min	30 Min	60 Min	90 Min	120 Min
Normal control	95 ± 3.9	178 ± 4.6	162 ± 4.1	130 ± 1.9	96 ± 4.7
DM control	267 ± 5.5	348 ± 6.1 *	390 ± 6.8 **	361 ± 4.9 **	348 ± 5.8 *
DM + LU/DIO	249 ± 4.8	324 ± 4.3 *	277 ± 3.5 *	209 ± 5.1 *	135 ± 2.6 *
DM + SeNPs	270 ± 7.7	365 ± 5.5 **	311 ± 6.0 **	230 ± 2.4 *	192 ± 3.6 *
DM + GA/LU/DIO-SeNPs	254 ± 8.9	318 ± 5.7 **	272 ± 4.3 *	201 ± 7.2 *	117 ± 6.1 * ^#^
DM + GBl	261 ± 7.5	315 ± 5.4 *	293 ± 8.6 *	261 ± 6.9 *	170 ± 8.4 **

Results are expressed as mean ± SD. * *p* < 0.05 and, ** *p* < 0.01 vs. diabetic group (*n* = 8 for each group). Glibenclamide (GBl). Time after glucose load. ^#^
*p* < 0.05 vs. SeNPs group.

**Table 5 molecules-27-05642-t005:** Effect of LU/DIO, SeNPs GA/LU/DIO-SeNPs on insulin tolerance test (ITT) in diabetic mice.

Groups	Plasma Glucose (mg/dL)				
	0 Min	30 Min	60 Min	90 Min	120 Min
Normal control	91 ± 2.5	70 ± 1.7	72 ± 0.9	75 ± 2.4	76 ± 3.4
DM control	360 ± 3.8	270 ± 4.5 *	251 ± 1.9 **	249 ± 5.0 **	247 ± 4.2 **
DM + LU/DIO	370 ± 5.1	252 ± 3.4 **	216 ± 4.6 **	185 ± 5.5 *	154 ± 3.9 *
DM + SeNPs	390 ± 6.8	268 ± 4.7 *	222 ± 5.8 *	220 ± 6.2 **	215 ± 3.6 **
DM + GA/LU/DIO-SeNPs	293 ± 2.8	216 ± 6.7 ** ^#^	144 ± 3.2 * ^#^	129 ± 2.3 * ^#^	108 ± 5.0 * ^#^
DM + GBl	01 ± 4.1	248 ± 4.5 *	203 ± 3.3 *	193 ± 6.0 **	137 ± 2.1 ** ^#^

Results are expressed as mean ± SD. * *p* < 0.05 and, ** *p* < 0.01 vs. diabetic group (*n* = 8 for each group). Glibenclamide (GBl). Time after insulin injection. ^#^
*p* < 0.05 vs. SeNPs group.

**Table 6 molecules-27-05642-t006:** Effect of flavonoid formulation LU/DIO, SeNPs, and GA/LU/DIO-SeNPs on different serum, hepatic, and pancreatic markers in STZ-induced diabetic mice.

Parameters	10 mg/kg					
	ND	DM	DM + LU/DIO	DM + SeNPs	DM + GA/LU/DIO/SeNPs	DM + Mtf (150 g/kg)
SERUM						
Serum glucose (mmol/L)	5.0 ± 1.2	12.58 ± 4.1 ^a^	7.54 ± 2.7 *	8.53 ± 0.9 *	5.26 ± 0.7 ** ^#^	6.12 ± 1.0 ** ^#^
Serum insulin (mIU/L)	39.1 ± 3.46	32.5 ± 5.02 ^b^	37.5 ± 4.56 *	34.4 ± 3.5 *	38.8 ± 4.57 ** ^#^	37.2 ± 4.23 ** ^#^
HOMA-IR	8.7 ± 3.19	18.2 ± 1.98 ^a^	12.6 ± 3–75 *	13.4 ± 2.13 **	9.07 ± 2.78 * ^#^	10.11 ± 2.5 * ^#^
HbA1c (%)	5.3 ± 0.86	12.7 ± 2.60 ^c^	7.3 ± 1.74 **	8.2 ± 0.99 *	6.1 ± 0.85 ** ^#^	8.4 ± 1.23 *
HDL-CHO (mmol/L)	1.38 ± 0.12	0.61 ± 0.03 ^c^	0.90 ± 0.05 * ^#^	0.79 ± 0.02 **	0.99 ± 0.04 * ^#^	0.89 ± 0.01 ** ^#^
T-CHO (mmol/L)	1.82 ± 0.04	3.60 ± 0.62 ^b^	2.41 ± 0.36 **	2.68 ± 0.29 **	2.10 ± 0.26 * ^#^	2.12 ± 0.48 * ^#^
TG (mmol/L)	0.87 ± 0.05	1.59 ± 0.15 ^b^	1.18 ± 0.32 *	1.47 ± 0.27 **	0.98 ± 0.21 ** ^#^	1.01 ± 0.31 * ^#^
LDL (mmol/L)	111 ± 4.30	132 ± 5.01 ^a^	124 ± 3.28 **	127 ± 5.19 *	117 ± 5.21 ** ^#^	120 ± 4.43 * ^#^
NEFA (mg/dl)	3.6 ± 0.08	4.8 ± 0.07 ^b^	3.9 ± 0.06 **	4.1 ± 0.2 *	3.6 ± 0.05 ** ^#^	3.8 ± 0.08 **
LIVER						
Liver weight (g)	1.6 ± 0.08	2.2 ± 0.07 ^b^	1.5 ± 0.05 *	1.7 ± 0.08 **	1.4 ± 0.02 *	1.4 ± 0.05 *
MDA (µmol/L)	1.59 ± 0.07	4.61 ± 0.01 ^c^	1.60 ± 0.02 **	1.69 ± 0.03 **	1.53 ± 0.08 *	1.61 ± 0.04 *
SOD (U/mg protein)	294.2 ± 6.10	172 ± 4.53 ^a^	245 ± 6.40 ** ^#^	200.9 ± 6.74 **	288.0± 6.37 * ^#^	276.0 ± 4.39 * ^#^
CAT (U/mg protein)	50.23 ± 3.62	44.65 ± 2.56 ^b^	46.57 ± 5.12 *	47.64 ± 5–38 *	51.48 ± 2.97 **	49.81 ± 4.35 **
GSH-Px (µmol/L)	45.19 ± 1.49	36.52 ± 1.87 ^c^	42.35 ± 2.45 *	41.27 ± 4.23 **	46.51 ± 3.12 **	44.61 ± 3.19 **
AST or (Unit/mL)	20 ± 2.12	40 ± 3.49 ^b^	26 ± 1.87 **	29 ± 2.39 *	22 ± 1.89 * ^#^	24 ± 4.07 * ^#^
ALT (Unit/mL)	24 ± 1.98	44 ± 3.12 ^c^	28 ± 2.67 **	31 ± 4.01 **	24 ± 2.18 * ^#^	27 ± 4.12 * ^#^
ALP (Unit/mL)	99.32 ± 7.44	309.43 ± 6.38 ^a^	112.16 ± 7.44 ** ^#^	143.60 ± 6.39 *	100.35 ± 8.52 ** ^#^	108.97 ± 9.12 ** ^#^
Glycogen (mg/g) tissue)	15.2 ± 3.423	8.7 ± 2.16 ^b^	13.1 ± 4.08 ** ^#^	10.2 ± 3.06 *	14.7 ± 2.79 ** ^#^	12.8 ± 3.54 * ^#^
Kidney						
Kidney weight (g)	0.53 ± 0.06	0.52 ± 0.04 ^a^	0.54 ± 0.01	0.53 ± 0.04	0.52 ± 0.01	0.53 ± 0.04
MDA (µmol/L)	1.40 ± 0.07	7.44 ± 1.05 ^c^	1.56 ± 0.08 * ^#^	2.31 ± 0.09 *	1.43 ± 0.04 ** ^#^	1.69 ± 0.05 * ^#^
SOD (U/mg protein)	17.44 ± 1.35	8.12 ± 3.11 ^a^	17.79 ± 3.11 *	15.1 ± 3.11 *	19.6 ± 2.34 ** ^#^	18.12 ± 4.29 **
GSH-Px (U/mg protein)	31.12 ± 5.24	11.89 ± 3.48 ^c^	30.84 ± 5.06 ** ^#^	25.76 ± 1.89 *	32.29 ± 1.75 ** ^#^	25.42 ± 2.39 *
CAT (U/mg protein)	39.05 ± 3.64	10.72 ± 3.11 ^a^	36.65 ± 1.90 * ^#^	30.46 ± 2.51 **	40.02 ± 2.34 ** ^#^	33.13 ± 1.98 *

Normal control (ND), diabetic control (DM), thiobarbituric acid reactive products or malondialdehyde (MDA), superoxide dismutase (SOD), glutathione peroxidase (GPx), glutathione S-transferase (GST), aspartate aminotransferase (AST), alanine aminotransferase (ALT). Results are expressed as means ± SD (*n* = 8). * *p* < 0.05 and, ** *p* < 0.01 vs. diabetes group; ^a^
*p* < 0.05, ^b^
*p* < 0.01 and, **^c^**
*p* < 0.001 vs. control group (*n* = 8 for each group). ^#^
*p* < 0.05 vs. SeNPs group.

## Data Availability

The data that support the findings of this study are contained within the article.

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
