# Peer review of "Evaluation of Diabetes Effects of Selenium Nanoparticles Synthesized from a Mixture of Luteolin and Diosmin on Streptozotocin-Induced Type 2 Diabetes in Mice"

_molecules, 2022, doi:10.3390/molecules27175642_

Round 1
Reviewer 1 Report (Previous Reviewer 2)
The manuscript entitled "Evaluation of Diabetes effects of Selenium Nanoparticles Synthesized from a Mixture of Luteolin and Diosmin on Strepto- 3 zotocin-Induced Type 2 Diabetes in Mice" and submitted for publication is a revised version of the original, which incorporates some of the suggestions and answers for some of the questions raised earlier.
One of the main objections raised, the way the manuscript was written, has been significantly improved in this revised version, and is of acceptable quality. However, there are several misformatting throughout the text that need to be corrected (for example, in line 219). The tables also need to be carefully revised, improving their graphic appearance and readability.
Some of the questions that were not and need to be answered:
1) Why do diabetic animals have, on average, a higher weight than the other groups, at the beginning of the experiment (table 1)? The answer given by the authors in the comments was that they formed a group with the heaviest animals. This should not be done. The animals should be randomly assigned to the various groups at the beginning of the experiments.
2) The adminstration of the coumpounds should be also preformed in the control group and not only in the diabetic animals.
3) The authors changed, from the 1st version to the current one, the number of animals used and the way diabetes was induced. What was the reason for these changes? The authors should explain the changes.
4) Even though it now seems to me to be an adequate method, their description is still not correct. The authors say that they administered STZ 40 mg/ip. Don't they mean 40mg/kg, IP?
5) Why did they use metformin as a positive control in some experiments and Glibenclamide in others, whose results are shown in tables 2 and 3?
6) Why are the data regarding glucose levels recorded in control animals (table 2) different from those presented in the previous version of the manuscript (table 1), although the number of animals was the same?
8) The authors mention that there is a synergistic effect of selenium and flavonoids (for example in line 223, when referring to the ITT results). However, they did not perform a statistical analysis, comparing the results of the groups DM+SeNPs Vs. DM+ LU/DIO Vs. DM+ GA /LU/DIOSeNPs; without this analysis they cannot state that there is this synergistic effect, as they may not be statistically different.
9) The authors present new data (ITT) requested in my comments, in a short period of time. Anyone who works with animals knows that it is usually difficult to perform experiments on a significant number of animals in a short period of time. Moreover these experiments take 6 weeks. Either the authors already had this data and chose not to include it in the original version, or they performed the experiments in an impossible amount of time. The authors should justify this issue.
Author Response
The manuscript entitled "Evaluation of Diabetes effects of Selenium Nanoparticles Synthesized from a Mixture of Luteolin and Diosmin on Strepto- 3 zotocin-Induced Type 2 Diabetes in Mice" and submitted for publication is a revised version of the original, which incorporates some of the suggestions and answers for some of the questions raised earlier.
One of the main objections raised, the way the manuscript was written, has been significantly improved in this revised version, and is of acceptable quality. However, there are several misformatting throughout the text that need to be corrected (for example, in line 219).
Line 219 was corrected
The tables also need to be carefully revised, improving their graphic appearance and readability.
The tables were changed
Some of the questions that were not and need to be answered:
1) Why do diabetic animals have, on average, a higher weight than the other groups, at the beginning of the experiment (table 1)? The answer given by the authors in the comments was that they formed a group with the heaviest animals. This should not be done. The animals should be randomly assigned to the various groups at the beginning of the experiments.
Excuse me if I do not agree with your point of view, since I have informed you how the experiment was carried out, 144 adult male C57BL/6 mice (10 weeks of age) were taken, the groups were formed by weight ranges in such a way that the administration of the samples were equivalent since they were administered in mg/kg of weight.
We don't understand what the problem because we chose the heaviest animals to form a group. if they received a dose proportional to their weight
Possibly the problem about the number of animals is because we did not specify how many of them were used in each experiment.
48 animals for diabetes test
48 animals for OGTT
48 animals for ITT
Total= 144 animals
In the text each experiment the number of animals used was added
2) The adminstration of the coumpounds should be also preformed in the control group and not only in the diabetic animals.
Excuse me if I do not agree with your point of view, since testing all samples in normoglycemic animals is not part of the objective of this research
3) The authors changed, from the 1st version to the current one, the number of animals used and the way diabetes was induced. What was the reason for these changes? The authors should explain the changes.
There is no change in the number of animals used each group was 8 animals each and there were 6 groups The total number of animals was changed to 48 because in the previous version the 5 group was repeated, that's why it was the error which has already been corrected
Possibly the problem about the number of animals is because we did not specify how many of them were used in each experiment.
48 animals for diabetes test
48 animals for OGTT
48 animals for ITT
Total= 144 animals
4) Even though it now seems to me to be an adequate method, their description is still not correct. The authors say that they administered STZ 40 mg/ip. Don't they mean 40mg/kg, IP?
Mistake was corrected
5) Why did they use metformin as a positive control in some experiments and Glibenclamide in others, whose results are shown in tables 2 and 3?
Both oral hypoglycemic agents were used depending on the type of experiment since glibenclamide has better results in the in the OGTT and ITT than metformin and is the proposal for the type of experiments performed
ESERCH ARTICLE Open Access
Mohammad Faisal, Ahamed Ismail Hossain, Shahnaz Rahman, Rownak Jaha1 and Mohammed Rahmatullah. A preliminary report on oral glucose tolerance and antinociceptive activity tests conducted with methanol extract of Xanthosoma violaceum aerial Parts. BMC Complementary and Alternative Medicine 2014, 14:335. http://www.biomedcentral.com/1472-6882/14/335
6) Why are the data regarding glucose levels recorded in control animals (table 2) different from those presented in the previous version of the manuscript (table 1), although the number of animals was the same?
As I explain below, they are other diabetic animals different from those treated for 6 weeks Excuse me if I don't treat it right
In relation to the Oral Glucose Tolerance Test (OGTT) and Insulin tolerance test (ITT) on diabetic mice these are the methods used. We administrated 60 min before orally glucose dosed (2 g/Kg) to animals diabetic without any treatment (untreated for 6 weeks) with LU/DIO, SeNPs and GA
/LU/DIO-SeNPs and after blood was sampled at different time-points 30, 60, 90 and 120 min, glucose level was determined according to the techniques previously described.
4.11. Oral Glucose Tolerance Test (OGTT) on diabetic mice For the study of the acute anti-hyperglycemic effect of LU/DIO, SeNPs and GA /LU/DIO-SeNPs in 48 diabetic mice no treatment an oral glucose tolerance test was performed in fasted mice (6 h between 7:00 and 13:00 h prior to the assay, as suggested for C57BL/6 mice) [ 43]. were submitted to blood extraction, representing time 0. After were administered by oral gavage GA /LU/DIO-SeNPs (10 mg/Kg) and water (control), or reference compound glibenclamide (5 mg/Kg) 60 min before orally glucose dosed (2 g/Kg) and the and blood was sampled at different time-points 30, 60, 90 and 120 min, glucose level was determined according to the techniques previously described at
4.12. Insulin tolerance test (ITT)
The same fasting protocol used for OGTT was adopted for this test. Then 48 diabetic mice were injecting intraperitoneally with insulin (2 U/kg). The glucose level was determined according to the technique previously described at 15, 30, 60, 90 and 120 min after the insulin injection [44].
These methods in the manuscript were worded differently to improve understanding
8) The authors mention that there is a synergistic effect of selenium and flavonoids (for example in line 223, when referring to the ITT results). However, they did not perform a statistical analysis, comparing the results of the groups DM+SeNPs Vs. DM+ LU/DIO Vs. DM+ GA /LU/DIOSeNPs; without this analysis they cannot state that there is this synergistic effect, as they may not be statistically different.
The statistically different was performed in all experiments (LU/DIO and GA /LU/DIO-SeNPs) compared with SeNPs effects which confirms the synergistic effect of selenium, and flavonoids
9) The authors present new data (ITT) requested in my comments, in a short period of time. Anyone who works with animals knows that it is usually difficult to perform experiments on a significant number of animals in a short period of time. Moreover, these experiments take 6 weeks. Either the authors already had this data and chose not to include it in the original version, or they performed the experiments in an impossible amount of time. The authors should justify this issue
Hence the experiments to determine the Oral Glucose Tolerance Test (OGTT) and Insulin tolerance test (ITT) require 1 hour and after only 30, 60, 90 and 120 min (only is necessary one day the study) and not 6 weeks. In each experiment we used 48 different diabetic animals (untreated). We were able to do the ITT because fortunately we had diabetic animals in at that moment which another experiment was going to be carried out
Faisal, M.; Hossain, A. I.; Rahman, S.; Jaha, R.; Rahmatullah, M. A preliminary report on oral glucose tolerance and an tinociceptive activity tests conducted with methanol extract of Xanthosoma violaceum aerial Parts. BMC Complementary and Alternative Medicine 2014, 14:335. http://www.biomedcentral.com/1472-6882/14/335
Reviewer 2 Report (Previous Reviewer 1)
The manuscript may contain interesting results; however, it is not acceptable in the present form as it contains several typographical errors, the English form must be revised, but, above all, the manuscript reports scientifically relevant errors (e.g., lines 78 and 79, the authors report that "research has shown that LU increases insulin and glucose levels"!); the authors include Glutathione in antioxidant enzymes, but GSH is not an enzyme! On page 3, line 117 the authors comment on fig. 3 (A-C), but, in reality, it should be fig. 2.
More, in many cases the abbreviation is given several lines before indicating what it refers to (see lines 134 and 138). Regarding in vivo experiments, the description of some methods is inaccurate; in addition, the authors report that the study was conducted for 6 weeks, but the determination of glycated hemoglobin gives valid information and must be carried out after 7-12 weeks; Table 1 is incomprehensible: what does "parame" mean? the formatting of the table makes it difficult to understand whether the values in the columns refer to weight, food intake or water intake; again, were these parameters measured only on the first day and after 6 weeks? In the materials and methods section the authors claim to have measured the levels of adiponectin and leptin in serum, but in the manuscript, there are no related results.
In summary, the work is confusing, full of results not well commented on, and the conclusions are hasty and summary.
The manuscript is not acceptable
Author Response
The manuscript may contain interesting results; however, it is not acceptable in the present form as it contains several typographical errors, the English form must be revised, but, above all, the manuscript reports scientifically relevant errors (e.g., lines 78 and 79, the authors report that "research has shown that LU increases insulin and glucose levels"!); the authors include Glutathione in antioxidant enzymes, but GSH is not an enzyme!
In the article we included glutathione peroxidase (GSH-Px or GPx )which is an enzyme. GSH was no included
Katar M. Ozugurlu A.F. Ozyurt H. Benli I. (2014). Evaluation of glutathione peroxidase and superoxide dismutase enzyme polymorphisms in celiac disease patients. Genet. Mol. Res. 13(1): gmr2956. https://doi.org/10.4238/2014.February.20.4
Danish Ahmed1, Vikas Kumar, Amita Verma, Pushpraj S Gupta, Hemant Kumar, Vishal Dhingra,
Vatsala Mishra and Manju Sharma, Antidiabetic, renal/hepatic/pancreas/cardiac protective and antioxidant potential of methanol/dichloromethane extract of Albizzia Lebbeck Benth. stem bark (ALEx) on streptozotocin induced diabetic rats. BMC Complementary and Alternative Medicine 2014, 14:243. http://www.biomedcentral.com/1472-6882/14/243
lines 78 and 79, the authors report that "research has shown that LU increases insulin and glucose levels"!); On page 3, line 117 the authors comment on fig. 3 (A-C), but, in reality, it should be fig. 2.
These mistakes were corrected
More, in many cases the abbreviation is given several lines before indicating what it refers to (see lines 134 and 138).
These mistakes were corrected
Regarding in vivo experiments, the description of some methods is inaccurate; in addition, the authors report that the study was conducted for 6 weeks, but the determination of glycated hemoglobin gives valid information and must be carried out after 7-12 weeks;
In the following article glycated hemoglobin was determined after 4 weeks of treatment
Zhou D, Chen L, Mou X. Acarbose ameliorates spontaneous type-2 diabetes in db/db mice by inhibiting PDX-1 methylation. Mol Med Rep. 2021, 23(1):72. doi: 10.3892/mmr.2020.11710.
In other study did not find a significant variation when increasing the weeks. The mean HbA1c levels at weeks 0, 4, and 12 were 7.7%, 7.2%, and 7.1%
Lee Y-M, Kim S-A, Lee I-K, Kim J-G, Park K-G, Jeong J-Y, et al. (2016) Effect of a Brown Rice Based Vegan Diet and Conventional Diabetic Diet on Glycemic Control of Patients with Type 2 Diabetes: A 4-12-Week Randomized Clinical Trial. PLoS ONE 11 (6): e0155918. doi: 10.1371/journal.pone.0155918
In other study glycated hemoglobin was determined 5 weeks
Kuiniu Zhu, Zhaoqing Meng, Yushan Tian, Rui Gu a, Zhongkun Xu a, Hui Fang, Wenjun Liu a, Wenzhe Huang a, Gang Ding a, Wei Xiao Hypoglycemic and hypolipidemic effects of total glycosides of Cistanche tubulosa in diet/streptozotocin-induced diabetic rats. Journal of Ethnopharmacology 276 (2021) 113991
Table 1 is incomprehensible: what does "parame" mean? the formatting of the table makes it difficult to understand whether the values in the columns refer to weight, food intake or water intake; again, were these parameters measured only on the first day and after 6 weeks?
Table 1 was divided into 3 tables as following:
Table 1. Effect of flavonoid formulation (LU/DIO), SeNPs, and GA /LU/DIO-SeNPs on weight, in diabetic mice (0 days, 1 weeks, 2 weeks, 3 weeks, 4 weeks, 5 weeks, and 6 weeks). Table 2. Effect of flavonoid formulation (LU/DIO), SeNPs, and GA /LU/DIO-SeNPs on water intake in diabetic mice (0 days, 1 weeks, 2 weeks, 3 weeks, 4 weeks, 5 weeks, and 6 weeks). Table 3. Effect of flavonoid formulation (LU/DIO), SeNPs, and GA /LU/DIO-SeNPs on food intake, in diabetic mice (0 days, 1 weeks, 2 weeks, 3 weeks, 4 weeks, 5 weeks, and 6 weeks).
In the materials and methods section the authors claim to have measured the levels of adiponectin and leptin in serum, but in the manuscript, there are no related results.
Adiponectin and leptin in serum were not evaluated in this study the error was corrected
In summary, the work is confusing, full of results not well commented on, and the conclusions are hasty and summary. The manuscript is not acceptable
The abstract and conclusions were revised
This manuscript is a resubmission of an earlier submission. The following is a list of the peer review reports and author responses from that submission.
Round 1
Reviewer 1 Report
The subject of the manuscript is interesting, but it is inadequately treated; above all, the part that should deal with the execution of the experiments in vivo is lacking: there is no trace of the modalities of induction of diabetes by streptozotocin and the authors argue that there were no significant differences in the intake of fluids and food in diabetic animals, but it is known, instead, that the condition of hyperglycemia profoundly modifies these (and not only these) two parameters. The authors talk about 5 weeks, but then in the table they report the results after 45 days; some sentences and concepts need to be revised (e.g. line 37, Introduction, are ROS caused by oxidative stress? ROS causes oxidative stress!); line 46 it is not clear what the authors mean.
Reviewer 2 Report
The theme presented in the manuscript entitled "Evaluation of Diabetes effects of Selenium Nanoparticles Synthesized from a Mixture of Luteolin and Diosmin on Streptozotocin-Induced Type 2 Diabetes in Mice", submitted for publication in the journal Molecules is interesting and current. In fact, much attention has been recently given to the use of bioactive compounds in the prevention and treatment of many diseases and in particular diabetes.
However, the presented work presents a number of weaknesses that need to be corrected and/or better developed, so that the work can be accepted for publication.
Thus:
1) The manuscript is written in an unclear way, which makes it very difficult to read. The authors should make an effort to make the text clearer, more fluid, so that readers can easily understand the message they intend to transmit.
2) I don't understand how the statistical analysis was performed. The authors mention in the figure captions that "Values with different letters are significantly different from each other at p < 0.05 as determined by Turkey's multiple-range test". What does this mean? For example, in table 3 in the row referring to serum glucose levels, we have values that are accompanied with the letters a, c or b and values without letters. What does this mean? Which values are different from each other? The authors should clarify the analysis.
3) In this paper, the authors used mice injected with STZ as an animal model of type 2 diabetes. In most studies with this type of animal model, diabetes is induced by multiple injections of STZ (between 3 and 5) of small doses of STZ. How can they guarantee that the model is actually type 2 diabetes and not type 1 diabetes? In addition to calculating the HOMA-Ir, the authors should have performed an insulin tolerance test on the animals.
4) The authors mention that they used 35 animals, divided into 7 groups. This allows to obtain 5 animals/group. However, the authors refer in the legends of the tables an n=8. How was this n obtained? Were the animals measured more than once?
5) Why do diabetic animals have, on average, a higher weight than the other groups, at the beginning of the experiment (table 2)?
6) In the legends of the tables, the authors state that "Results are expressed as means ±DS". Ds or SD (standard deviation)?
7) In some results the authors chose to present the variations as percentages (%) and in others as fold increase. The authors should standardize the way they present these variations, thus facilitating the analysis of the results.
8) The results presented in table 1 were obtained with diabetic animals (as mentioned in the title of the table) or with control animals? The glucose values at time 0 point to control, non-diabetic animals. This analysis should also be done with diabetic animals.
9) The analysis of the results is hampered by what has already been explained in point 2. However, trying to analyze the results presented in table 3, it seems that the administration of empty selenium nanoparticles to diabetic animals (Se-NPs), without the bioactive compounds, reversed many of the changes caused by diabetes. How do the authors explain this effect? Did the release of LU and DIO by the nanoparticles have any significant additive effect compared to the addition of just the nanoparticles? From the analysis of the table I see some differences but I don't know if they are statistically significant.
10) Table 3 needs to be formatted, because the way they are separated gives the idea that DM+GA/LUDIO-SeNPs were administered at a dose of 150mg/kg. It should also be indicated that the administration was daily (xmg/kg/day).
11) There are numerous textual misformatting throughout the manuscript. Authors should carefully review the entire text, correcting these problems.
In conclusion, although the work is interesting, there are some problems, from issues of the way the text is written to scientific and methodological issues, that need to be corrected before the manuscript can be accepted for publication.